# Theoretical Study on the Influence of the Anharmonic Effect on the Ionic Conductivity and Thermal Stability of 8 mol% Yttria-Stabilized Zirconia Solid Electrolyte Material

**DOI:** 10.3390/ma16155345

**Published:** 2023-07-29

**Authors:** Junhua Gao, Xiaofeng Zhao, Zhengfu Cheng, Liangliang Tian

**Affiliations:** 1Key Laboratory of Electronics Engineering, Heilongjiang University, Harbin 150080, China; 1202879@s.hlju.edu.cn; 2School of Electronic Information and Electrical Engineering, Chongqing University of Arts and Sciences, Chongqing 402160, China

**Keywords:** 8 mol% YSZ solid electrolyte, ionic conductivity, anharmonic effect, thermal stability

## Abstract

YSZ is a promising material for resistive memory devices due to its high concentration of oxygen vacancies, which provide the high anion migration rates crucial for the manifestation of resistance switching in metal oxides. Therefore, investigating the ionic conductivity of YSZ is an important issue. The ionic conductivity and thermal stability of 8 mol% YSZ were studied using the theories and methods of solid-state physics and physical chemistry. The impact of anomalous atomic vibrations on the material was also explored, and the variation in the ion vibration frequency, electrical conductivity, and thermal stability coefficient of electrical conductivity with temperature was obtained. The results show that the ion conductivity of an 8 mol% YSZ solid electrolyte increases nonlinearly with temperature, with a smaller increase at lower temperatures and a larger increase at higher temperatures. Considering the anharmonic effect of ion vibrations, the electrolyte conductivity is higher than the result of the harmonic approximation, and the anharmonic effect becomes more significant at higher temperatures. Our research fills the gap in the current literature regarding the theoretical non-harmonic exploration of the ion conductivity and thermal stability factor of YSZ solid electrolytes. These results provide valuable theoretical guidance for the development and application of high-performance YSZ resistive memory devices in high-temperature environments.

## 1. Introduction

YSZ (yttria-stabilized zirconia) resistive memory has great potential for use in high-temperature environments, such as the aerospace, automotive, energy, and petrochemical industries [1,2]. In these fields, electronic devices operating in high-temperature environments require high-temperature stability and reliability. However, the thermodynamic properties of solid electrolytes, such as thermal expansion, can significantly impact their conductivity. Therefore, several studies have been conducted to investigate the electrical conductivity and stability of solid electrolyte materials. In terms of experimental research, Kwon [3], Kumar [4], Park [5], Borik [6] et al. prepared doped ZrO_2_ solid electrolyte materials and measured the electrical conductivity of the prepared materials by electrical impedance spectroscopy, open-circuit potential measurement, and other methods, and they studied the effects of the defect structure, doping concentration, temperature, and other factors on the electrical conductivity of the material. In terms of theoretical research, Lima [7] studied the thermal stability of yttria-stabilized zirconia coatings obtained by plasma spraying. Glinchuk [8] studied the effect of surface tension on the activation energy of oxygen ion conduction in nanodynamics and calculated the activation energy of oxygen ion diffusion through oxygen vacancies. Zavodinsky [9,10] used the electron density functional method (gradient approximation) and pseudopotential method to study the conduction mechanism of the ionic conductivity of ZrO_2_ solid electrolytes and the stability mechanism of cubic zirconia and zirconia nanoparticles with different stoichiometries. Hirschorn [11] introduced a comprehensive model for predicting and explaining the ionic conductivity of YSZ solid electrolytes. Song [12] discussed the effects of temperature and oxygen partial pressure on the electrical conductivity of YSZ electrolytes. Etschmann [13] presented a molecular dynamics study of the ionic conductivity of YSZ. Wang [14], Mahapatra [15], and Yildiz [16] studied the effects of the microstructure, grain size, and grain boundary on YSZ ionic conductivity. These references cover various aspects of YSZ ionic conductivity, including the effects of temperature, oxygen partial pressure, microstructure, and other factors. These studies are of great significance for understanding the electrochemical properties of YSZ and its application in high-temperature environments, such as ion batteries or solid oxide fuel cells.

Currently, research on the conductivity of ZrO_2_ solid electrolytes mainly relies on experimental methods. However, these methods cannot provide an analytical formula for the conductivity of solid electrolyte materials, making it difficult to theoretically reveal the mechanism of conductivity changes and other conductivity properties. Although some studies have theoretically investigated the ionic conductivity of YSZ materials, they did not consider the influence of anomalous atomic vibrations, and there are certain gaps between the results obtained and actual measurements. Moreover, the variations in the ionic conductivity and thermal stability of YSZ solid electrolyte materials with temperature have not been extensively studied. In this paper, we aim to apply solid-state physics theory to study the temperature dependence of the conductivity of 8 mol% YSZ solid electrolyte and explore the influence of anharmonic effects to provide valuable theoretical guidance for the development and application of high-performance YSZ resistive memory devices in high-temperature environments. Our research can fill the gap in the current literature regarding the theoretical non-harmonic exploration of the ion conductivity and thermal stability factor of YSZ solid electrolytes. Furthermore, it can provide a theoretical basis for the application research of YSZ in solid oxide fuel cells, oxygen sensors, thermal barrier coatings, and other related fields.

## 2. Methods

### 2.1. Physical Model

From a mesoscopic perspective, the solid electrolyte can be viewed as a cylindrical thin film made up of YSZ particles with a larger particle size and a thickness of *L*, as shown in Figure 1, in contact with the electrodes. YSZ has a cubic crystal structure [3]. For computational simplicity, it is assumed that in 8 mol% YSZ, the ZrO_2_ lattice consists of a complex lattice of cubic cells with a lattice constant of *a*, where the O atoms are located at the centers of 8 vertices and 6 faces, and the Zr atoms are located at the midpoints of 12 edges. The doped Y atoms replace the Zr atoms located at the body center. The crystal structures of undoped ZrO_2_ and Y-doped ZrO_2_ cells are shown in Figure 2a,b, respectively.

In the model, atoms are treated as particles that interact with each other and undergo anharmonic vibrations near their equilibrium positions. It is assumed that the distance between the two nearest atoms is *r*, and the average interaction energy is *u*(*r*), which expands near the equilibrium position *r*_0_. When deviating from the equilibrium position, represented by *ξ* = *r* − *r*_0_, its value is small:(1)$$u(r)=u(r_{0})+\frac{1}{2}b_{0}\xi^{2}+b_{1}\xi^{3}+b_{2}\xi^{4}+\ldots$$
where *ε*_0_ is the simple harmonic coefficient, and *ε*_1_ and *ε*_2_ are the first and second anharmonic coefficients, respectively.

For transition metals such as zirconium, the interaction potential between Zr atoms [17] is
(2)$$u_{Zr-Zr}(r)=\frac{g}{r}\exp(\frac{r}{r_{0}})\{\exp[\frac{-2n(r-r_{0})}{\lambda_{D}}]-2\exp[\frac{-n(r-r_{0})}{\lambda_{D}}]\}$$
where *λ_D_* = 4(π*F_C_*/6)*a* is the average Debye wavelength, *a* is the lattice constant, *F_C_* = 1/4 represents the structure factor for the face-centered cubic crystal, *g* is related to bonding, *n* is the bond strength parameter, and *r*_0_ is the equilibrium distance between the two nearest atoms, and the parameter values of Zr and Y [18,19] are seen in Table 1.

For Zr-O ionic bonding, the Zr-O atomic interaction potential [18] is:(3)$$u_{\mathrm{Zr}-O}(r)=z_{0}\lambda \exp(-\frac{r}{\rho })-\frac{\alpha q^{2}}{4\pi \varepsilon_{0}r}$$
where *q* represents the charge of the zirconium ion, and *α* = 1.6381 is the Madelung constant. $z_{0}$ represents the number of ions in the nearest neighbor of the Zr ion, while *ρ* represents the repulsive interaction range parameter, which can be set at 5 times the radius of the Zr ion. *λ* represents the repulsive energy parameter, with a value of *λ*_Z0_ = 2.1 × 10^−8^ erg [18]. The specific values of these parameters are provided in Table 2.

The interaction potential for O-O can be written in the form of Van der Waals potential [21]:(4)$$u_{O-O}(r)=\frac{D}{\left(b-a\right)}[a(\frac{r_{0}}{r}{)}^{b}-b{\left(\frac{r_{0}}{r}\right)}^{a}]$$
where *D* is the well depth, and *r*_0_ is the distance between two atoms when *u*(*r*) = 0. *a*, b, *r*_0_, and *D* can be determined by the specific potential energy curve.

Based on the structure shown in Figure 2 and Equations (2)–(4), the values of *ε*_0_, *ε*_1_, and *ε*_2_ were obtained by considering the nearest neighbor, second nearest neighbor, and next nearest neighbor, as shown in Table 3.

### 2.2. Conduction Mechanism of YSZ Electrolyte

ZrO_2_ is an ionic conductor crystal that conducts electricity through oxygen ion vacancies in its crystal lattice. Oxygen molecules are adsorbed on the electrode surface and transform into oxygen atoms. These atoms then diffuse to the three-phase interface on the electrode surface, where they become oxygen ions due to electrochemical reactions. These ions enter the YSZ electrolyte, migrate, and diffuse in the YSZ electrolyte along the path shown in Figure 3, before moving to the other electrode through the YSZ electrolyte to complete the circuit, thereby forming an electrical current. The addition of oxides generates a large number of oxygen ion vacancies in ZrO_2_. For Y_2_O_3_, every time two Y ions are added, one oxygen ion vacancy is generated. The concentration of oxygen ion vacancies mainly depends on the composition and quantity of the dopant [22].

The chemical reaction that occurs in the ZrO_2_ solid electrolyte with Y_2_O_3_ as a stabilizer can be expressed as [23]:(5)$$(1-x){\mathrm{ZrO}}_{2}+\frac{x}{2}Y_{2}O_{3}\to {\mathrm{Zr}}_{1-x}^{4+}+Y_{x}^{3+}+O_{2-\frac{x}{2}}^{2-}+\frac{x}{2}{\;V_{O}}^{\prime \prime }$$

The formation of oxygen ion vacancies can be described by the Kroger-Yink mechanism:(6)$$Y_{2}O_{3}+2\mathrm{Zr}+O^{-}\vec{\leftarrow }2Y_{\mathrm{Zr}}+{\;V_{O}}^{\prime \prime }+2{\mathrm{ZrO}}_{2}$$
where O^2−^ represents the oxygen ion in the lattice, V_O_^″^ represents the oxygen ion vacancy, Zr^4+^ represents the zirconium ion, and Y^3+^ represents the yttrium ion.

In addition to O^2−^, Zr^4+^, and Y^3+^, the charged particles in the electrolyte, electrons and holes *h*, also contribute to its conductivity. Assuming that the conductivities are *σ*O^2−^, *σ_e_*, and σ*_h_*, respectively, the total electrical conductivity of the electrolyte can be expressed as *σ* = *σ*O^2−^ + *σ_e_* + *σ_h_*. However, at temperatures below 1200 K, only oxygen ion conduction plays a major role. Therefore, the total electrical conductivity can be simplified to *σ* = *σ*O^2−^.

### 2.3. Ionic Conductivity in YSZ

The conduction of solid electrolytes is primarily ionic conduction, which results from the migration and diffusion of oxygen ions. As mentioned earlier, the addition of oxides generates a significant number of oxygen ion vacancies in ZrO_2_. Due to thermal excitation, when the ion energy exceeds the potential barrier between adjacent ions (referred to as the diffusion activation energy), the ion transitions from its original equilibrium position to the adjacent vacancy, creating vacancies at the original position. When the ions migrate and diffuse to the right, the vacancies migrate in the opposite direction.

Using the theory of solid-state physics and the vacancy diffusion ion conduction mechanism, the migration speed of oxygen ions along the direction of the electric field can be calculated [23]:(7)$$v_{d}=2\nu_{0}de^{-\varepsilon /k_{B}T}\sinh(\frac{Eqd}{2k_{B}T})$$

For general sensors, the electric field strength *E* involved is not large, *Eqd* ≪ *k*_B_*T*:(8)$$v_{d}=\frac{q}{k_{B}T}\nu_{0}d^{2}e^{-\varepsilon /k_{B}T}E=\mu E$$
where *μ* is the mobility, and its relationship with temperature is obtained from Equation (8) as
(9)$$\mu =\frac{\gamma q}{k_{B}T}d^{2}\nu_{0}\exp\left(-\frac{\varepsilon }{k_{B}T}\right)$$
where *γ* represents the geometric factor, *ν*_0_ represents the vibrational frequency of the vacancy (i.e., the vibrational frequency of the ion), *d* represents the transition distance, which is approximately equal to the lattice constant *a*, *q* represents the charge of the oxygen ion, and *ε* represents the change in the Gibbs function caused by the vacancy transition, i.e., the diffusion activation energy.

Considering the ion’s anomalous vibration, the relationship between the vibration frequency of the ion and the temperature [23] is
(10)$$\nu_{0}(T)=\nu_{0}[1+(\frac{15\varepsilon_{1}^{2}}{2\varepsilon_{0}^{3}}-\frac{2\varepsilon_{2}}{\varepsilon_{0}^{2}})k_{B}T]$$

Assuming that the number of oxygen ions per unit volume is *C_i_*, the current density is obtained from the speed of the oxygen ion migration in the direction of the electric field using Equation (8):(11)$$j=C_{i}qv_{d}=\sigma E,$$

By substituting Equation (10) into Equation (9), and then into Equations (8) and (11), and considering only the oxygen ion conductivity, the relationship between the electrolyte conductivity and temperature can be expressed as
(12)$$\sigma =\frac{q_{i}^{2}}{k_{B}T}C_{i}a\nu_{0}[1+(\frac{15\varepsilon_{1}^{2}}{2\varepsilon_{0}^{3}}-\frac{2\varepsilon_{2}}{\varepsilon_{0}^{2}})k_{B}T]\exp\left(-\frac{\varepsilon }{k_{B}T}\right)$$

The thermal expansion coefficient is a physical quantity that describes the proportional change in the length, area, or volume of a material with temperature. It is a measure of the deformation of the material during thermal expansion, indicating the ratio between the change in length, area, or volume and the original length, area, or volume. The thermal expansion coefficient is an important property in materials, especially at high temperatures, as the coefficient increases with temperature and may cause the material to deform or crack under heat. Therefore, for materials and devices that operate at high temperatures, controlling and optimizing the thermal expansion coefficient is critical. The thermal stability of the material is often described by the temperature stability coefficient, and the temperature stability coefficient *α_σ_* of the conductivity of the electrolyte material is defined as [24]
(13)$$\alpha_{\sigma }=\frac{1}{\sigma }(\frac{d\sigma }{dT})$$

## 3. Results and Discussion

In ref. [25], the masses of O, Zr, and Y atoms are reported as M_O_ = 0.265686 × 10^−25^ kg, M_Zr_ = 1.514811 × 10^−25^ kg, and M_Y_ = 1.476357 × 10^−25^ kg, respectively. Furthermore, the vibration frequencies at absolute zero, ω_0O_ = 9.375247 × 10^13^ s^−1^, ω_0zr_ = 7.853824 × 10^13^ s^−1^, and ω_0Y_ = 6.198625 × 10^13^ s ^−1^, of O, Zr, and Y, respectively, are obtained from *ω* = (ε_0_/M)^1/2^. By substituting the above data into Equation (10), the change in the oxygen ion vibration frequency *ν* = *ω*/2π with temperature is obtained, as shown in Figure 4. Curves 0, 1, and 2 represent the results of simple harmonic approximations considering only the first anharmonic term and considering both the first and second anharmonic terms, respectively. The same notation is used below.

It can be seen that the vibration frequency of oxygen ions remains constant in the simple harmonic approximation. However, after considering the anharmonic effect of ions, the vibration frequency of oxygen ions increases linearly with increasing temperature. The vibration frequency of only the first anharmonic term is greater than that of considering both the first and second anharmonic terms, and it is also greater than that of the simple harmonic approximation. Furthermore, the higher the temperature, the greater the difference between the vibration frequencies, indicating a more significant anharmonic effect. The influence of the anharmonic effect on the vibrational frequency of oxygen ions increases with increasing temperature. For example, when *T* is 700 K, the influence of the anharmonic effect on the vibrational frequency of the oxygen ion is 2.14%, while when *T* is 1200 K, it is 3.62%.

The anharmonic effect arises due to the nonlinearity of the interatomic potential energy function. In the simple harmonic approximation, the potential energy function is approximated as a symmetric parabolic well, and the motion of ions is assumed to be harmonic. This approximation works well at low temperatures when the amplitude of motion is small. However, at higher temperatures, the amplitude of motion is larger, and the potential energy function becomes asymmetric, leading to anharmonic behavior. The anharmonicity causes the interatomic potential energy function to deviate from the parabolic well, resulting in the variation in the vibrational frequency with increasing temperature. The anharmonic effect is more significant at higher temperatures, leading to a larger deviation from the simple harmonic approximation. Therefore, at high temperatures, the anharmonic effect becomes more pronounced, leading to an increase in the vibrational frequency of oxygen ions.

Ref. [6] provides the activation energy of oxygen ions doped with 8 mol% YSZ. In the temperature range of 300~1500 K, the lattice constant of the YSZ electrolyte with a doping concentration of 8 mol% varies with temperature approximately as *a* = (5.1218 + 4.2056 × 10^−5^ *T*) Å, as reported in ref. [26]. The quantity of oxygen ions is *q* = 2*e*, and the number of oxygen ions per unit volume is *C_i_* = 2.208 × 10^27^. By substituting the above data into Equation (12), the change in the electrolyte’s conductivity with temperature and the conductivity curve of the YSZ crystal with a doping concentration of 8 mol% given in ref. [26] are shown in Figure 5a. For comparison, the research results of ref. [27] are presented in Figure 5b. The consistency of the results in this paper with the changes in YSZ electrolyte conductivity with temperature reported in other documents, as indicated by refs. [26,27], further supports the validity of the calculations in this paper.

The conductivity of 8 mol% YSZ increases nonlinearly with increasing temperature. The conductivity is low when *T* is low, but it increases rapidly with temperature when *T* > 1000 K. For instance, when 800 K < *T* < 1000 K, the conductivity changes from 0.0024 S∙cm^−1^ to 0.025 S∙cm^−1^, which is an increase of 0.0226 S∙cm^−1^. When 1000 K < *T* < 1200 K, it changes from 0.025 S∙cm^−1^ to 0.115 S∙cm^−1^, which is an increase of 0.090 S∙cm^−1^. After taking into account the anharmonic effect of ion vibrations, the conductivity of the electrolyte is greater than that of the harmonic approximation. Moreover, the higher the temperature, the greater the difference in conductivity between the anharmonic and harmonic approximations. This indicates that the anharmonic effect becomes more significant at higher temperatures. When *T* is 1173 K, the conductivity of 8 mol% YSZ electrolyte is 0.096 S∙cm^−1^ in this paper, while it is 0.063 S∙cm^−1^ in ref. [6].

The conductivity of a solid electrolyte is related to the mobility of ions, which increases with temperature due to increased thermal energy. However, the conductivity does not increase linearly with temperature, and it may exhibit nonlinear behavior due to various factors, such as the concentration of defects, the crystal structure, and the anharmonic effect of ion vibrations. In the case of 8 mol% YSZ, the conductivity is low at low temperatures, but it increases rapidly with temperature when *T* > 1000 K. This nonlinear behavior can be attributed to the increased mobility of ions at higher temperatures, which leads to an increase in conductivity. However, the anharmonic effect of ion vibrations can also contribute to this behavior. As the temperature increases, the anharmonic effect becomes more significant, causing the ion vibrations to deviate from the harmonic approximation. This leads to an increase in conductivity compared to the harmonic approximation. The difference in conductivity between the anharmonic and harmonic approximations becomes greater at higher temperatures, indicating that the anharmonic effect becomes more significant at higher temperatures. Therefore, taking into account the anharmonic effect of ion vibrations is necessary for accurate predictions of conductivity at high temperatures.

Substituting Equation (12) into Equation (13), the variation curve of the conductivity’s temperature stability coefficient of the YSZ electrolyte with temperature can be obtained, as shown in Figure 6.

The temperature stability coefficient *α_σ_* of YSZ electrolyte conductivity decreases nonlinearly with increasing temperature. As the temperature continues to rise, *α_σ_* tends to be constant, indicating that the higher the temperature, the smaller the temperature stability coefficient of the conductivity. This suggests that the thermal stability of the conductivity of the YSZ electrolyte material is better. For example, when the temperature is 800 K < *T* < 1000 K and 1000 K < *T* < 1200 K, the temperature stability coefficient of electrolyte conductivity decreases by 52.6% and 31.5%, respectively. The harmonic approximation is consistent with the temperature stability coefficient of the electrolyte conductivity when the anharmonic effect is considered, indicating that the anharmonic effect has little effect on the temperature stability of the conductivity.

The temperature stability coefficient of YSZ electrolyte conductivity is a measure of how much the conductivity changes with temperature. A small temperature stability coefficient indicates that the conductivity is less sensitive to temperature changes, which is desirable for practical applications. In the case of YSZ electrolyte conductivity, the temperature stability coefficient decreases nonlinearly with increasing temperature. This behavior can be attributed to the fact that at higher temperatures, the mobility of ions increases, which leads to a larger change in conductivity with temperature. However, as the temperature continues to rise, the effect of increased ion mobility on conductivity becomes less significant, resulting in a more stable conductivity with respect to temperature. The decrease in the temperature stability coefficient of electrolyte conductivity with increasing temperature suggests that the thermal stability of the YSZ electrolyte material is better at higher temperatures. This is because the conductivity of the material changes less with temperature, making it more reliable for practical applications at high temperatures. The fact that the harmonic approximation is consistent with the temperature stability coefficient of the electrolyte conductivity when the anharmonic effect is considered indicates that the anharmonic effect has little effect on the temperature stability of the conductivity. This suggests that the anharmonic effect mainly affects the absolute value of the conductivity, rather than its temperature dependence.

## 4. Conclusions

This paper investigates the variation in ionic conductivity in 8 mol% YSZ solid electrolyte with temperature and the effect of the anharmonic vibration of ions by applying the theory and methods of solid-state physics. The results indicate the following: 1. The ionic conductivity of 8 mol% YSZ solid electrolyte increases nonlinearly with the increase in temperature. 2. The conductivity is low at low temperatures but increases rapidly with temperature when the temperature is higher than 1000 K. 3. After considering the anharmonic effect of ionic vibrations, the ionic conductivity is larger than the result of simple harmonic approximation, and the higher the temperature, the greater the difference between the anharmonic and simple harmonic approximations. This suggests that the anharmonic effect becomes more significant at higher temperatures. 4. These findings are consistent with other results in the literature on the variation in the conductivity of 8 mol% YSZ electrolyte with temperature, which demonstrates the validity of the calculation. By understanding the electrical conductivity of YSZ materials at different temperatures, researchers can better optimize the design and performance of YSZ-based resistive memory devices for specific applications. However, this paper also has some shortcomings: the studied YSZ solid electrolyte has a concentration of 8 mol%, and different concentrations of YSZ solid electrolytes may have different performance characteristics. So the results of this study may not be applicable to other concentrations of YSZ solid electrolytes. Furthermore, this study mainly focused on basic properties, such as ionic conductivity and thermal stability, and did not deeply investigate other properties relevant to practical application scenarios, such as reliability and durability. Therefore, a more comprehensive consideration is needed when applying these results to the design of practical devices.

## Figures and Tables

**Figure 1 materials-16-05345-f001:**
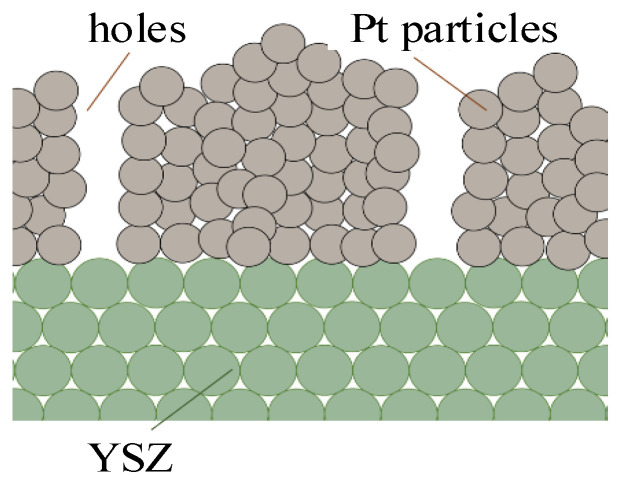
Schematic of electrodes and electrolytes near the three-phase interface.

**Figure 2 materials-16-05345-f002:**
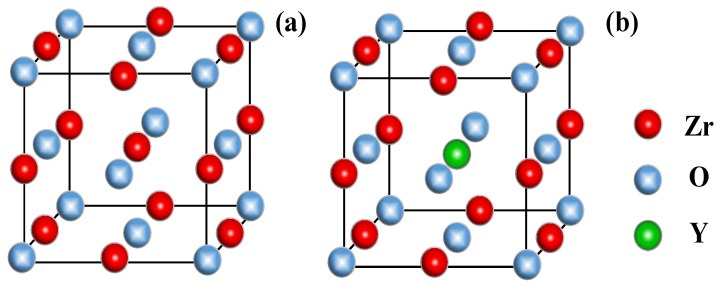
Schematic of undoped (**a**) and Y-doped (**b**) zirconia structure.

**Figure 3 materials-16-05345-f003:**
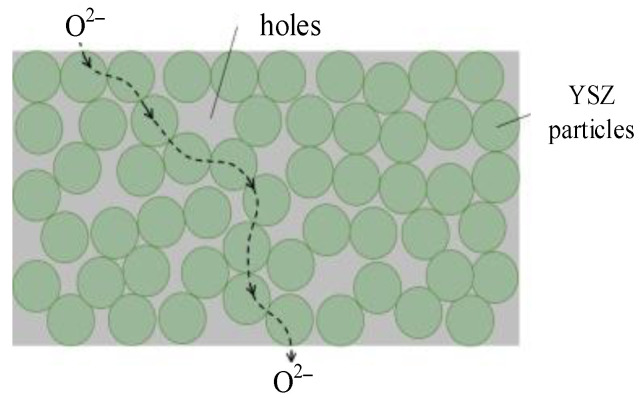
Diffusion model of oxygen ions in YSZ electrolyte.

**Figure 4 materials-16-05345-f004:**
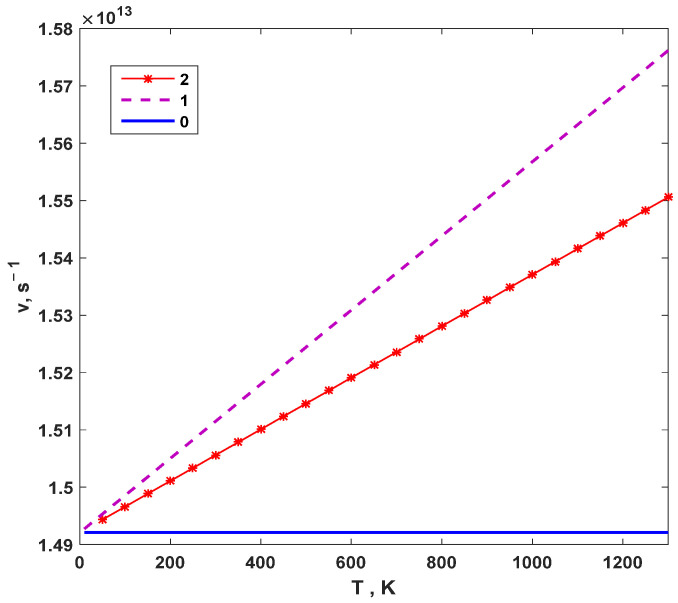
Variation in oxygen ion vibration frequency with temperature.

**Figure 5 materials-16-05345-f005:**
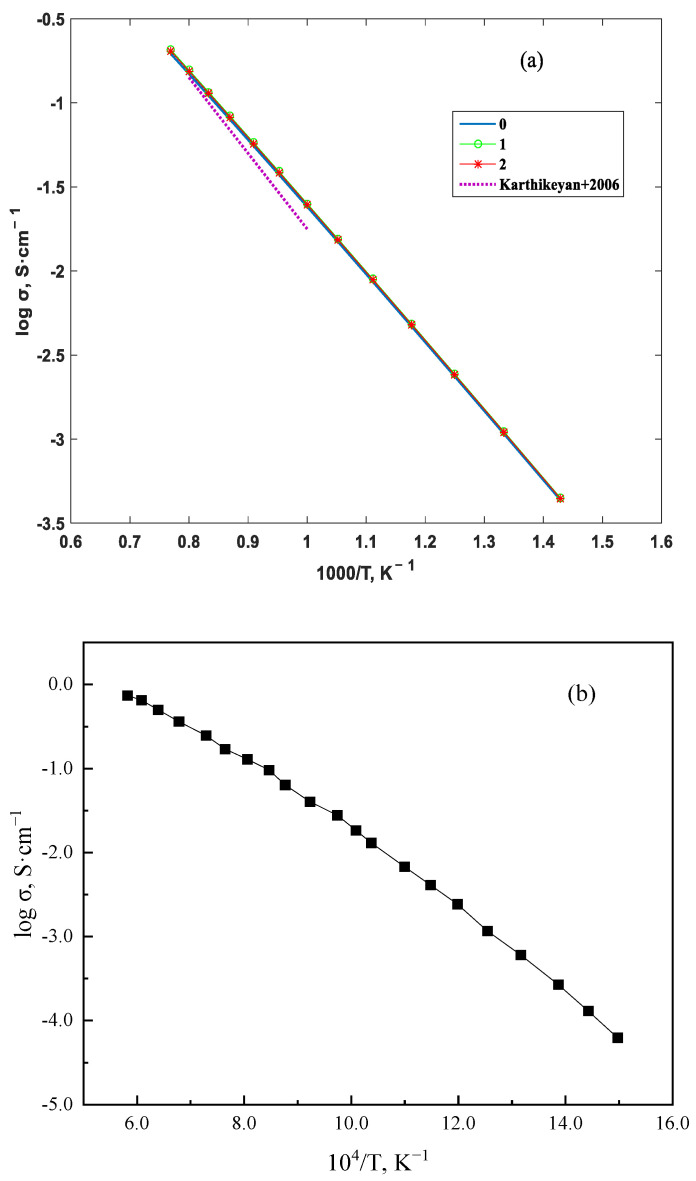
Variation in YSZ conductivity with temperature, and the dashed line represents the results of ref. [26] (**a**); research results of ref. [27] (**b**).

**Figure 6 materials-16-05345-f006:**
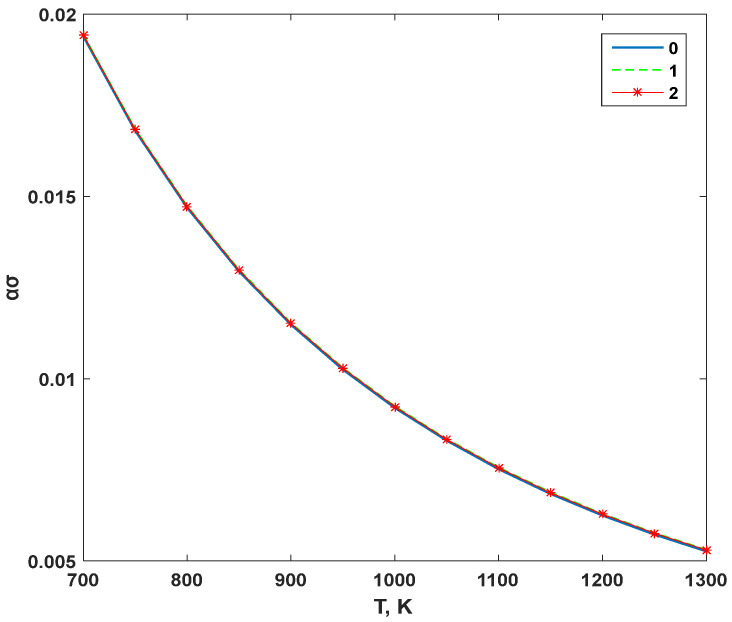
Variation in conductivity’s temperature stability coefficient with temperature.

**Table 1 materials-16-05345-t001:** Parameter values of Zr and Y.

	*a* (Å)	*r*_0_ (Å)	*λ*_d_ (Å)	*n*	g (eV∙Å)
Zr	3.23	3.27064	22.66438	1	75
Y	3.65	3.69592	25.61145	1	75

**Table 2 materials-16-05345-t002:** Born–Mayer–Buckingham potential parameters [18,19,20].

Ion Pairs	*A* (eV)	*ρ* (Å)	*C* (eV ∙ Å^6^)
O^2−^—O^2−^	9547.96	0.2192	32.0
Zr^4+^—O^2−^	1502.11	0.3477	5.1
Y^3+^—O^2−^	1766.40	0.3385	19.4

**Table 3 materials-16-05345-t003:** Simple harmonic, first anharmonic, and second anharmonic coefficients of each atom in YSZ.

	*ε*_0_ (10^2^ Jm ^−2^)	*ε*_1_ (10^12^ Jm ^−3^)	*ε*_2_ (10^22^ Jm ^−4^)
O	2.33525	−2.30901	2.60601
Zr	9.34374	−2.44726	2.37901
Y	8.00938	−0.39169	1.72522
YSZ	5.67260	−2.12119	2.41078

## Data Availability

Data sharing is not applicable to this article as no new data were created or analyzed in this study.

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
