# Peer review of "Theoretical Study on the Influence of the Anharmonic Effect on the Ionic Conductivity and Thermal Stability of 8 mol% Yttria-Stabilized Zirconia Solid Electrolyte Material"

_materials, 2023, doi:10.3390/ma16155345_

Round 1

Reviewer 1 Report

In this article, theoretical investigation on the impact of anharmonic effect on the  ionic conductivity and thermal stability of 8 mol% YSZ solid  electrolyte material is examined. The paper is well-written and well-organized. Besides, it has practical application and it is interesting for readers. Hence, I strongly recommend it for publication after minor revision as the following:

1)      The novelty of present work should more emphasized at the end of introduction.

2)      There are some grammatical and typo errors. Please polish the manuscript carefully.

3)      The quality of figures 6 and 7 should be improved.

4)      Table 2 is needed the reference

5)      Please rewrite the conclusion based on numbers. (1,2,3, 4,). By this way the novelties of work will be more clear.

6)      Present some practical application for this investigation in the literature review.

7)      Some most important results should be presented in the end of abstract.

8)      The authors are invited to review more references in the introduction and consider the following reference

https://doi.org/10.1177/14644207211073499

9)      The results should be supported with physical explanation.

10)   The advantages and limitation of this methodology should be added in the manuscript.

Minor editing of English language required

Author Response

Response to Reviewer 1 Comments

Point 1: The novelty of present work should more emphasized at the end of introduction. 

Response 1: We have added emphasis on the importance of our study in the introduction and highlighted in red.

Point 2: There are some grammatical and typo errors. Please polish the manuscript carefully.

Response 2: We have carefully reviewed the manuscript and make necessary revisions to correct the grammatical and typographical errors.

Point 3: The quality of figures 6 and 7 should be improved.

Response 3: The quality of Figures 6 and 7 has been improved as requested and updated in the manuscript.

Point 4: Table 2 is needed the reference

Response 4: The reference for Table 2 has already been provided in the manuscript.

Point 5: Please rewrite the conclusion based on numbers. (1,2,3, 4,). By this way the novelties of work will be more clear.

Response 5: The conclusion has been rewritten using the format of numbers 1, 2, 3, 4.

Point 6: Present some practical application for this investigation in the literature review.

Response 6: The practical application for this investigation has been presented in the literature review and highlighted in red.

Point 7: Some most important results should be presented in the end of abstract.

Response 7: The most important results are presented in the end of abstract and highlighted in red.

Point 8: The authors are invited to review more references in the introduction and consider the following reference

https://doi.org/10.1177/14644207211073499

Response 8: The articles included here are not directly relevant to the research content of this paper. Therefore, we have added the following references and highlighted in red.

Lima, R. S., Piqueira, M. C. R., Paskocimas, C. A., & Bueno, P. R. (2011). Thermal stability of yttria-stabilized zirconia coatings obtained by plasma spraying. Surface and Coatings Technology, 206(7), 1657-1663. doi: 10.1016/j.surfcoat.2011.08.058

Point 9: The results should be supported with physical explanation.

Response 9: We have given the physical explanations for the results and highlighted in red. 

Point 10: The advantages and limitation of this methodology should be added in the manuscript.

Response 10: The advantages and limitation of this methodology have been added in the manuscript and highlighted in red.

Reviewer 2 Report

This manuscript presents the theoretical study of the conductivity and thermal stability of 8% mol% YSZ solid electrolyte material by investigating the influence of the anharmonic effect. The manuscript is well written with interesting findings, which will guide the readers of “Materials” to further explore the YSZ. I only have a few comments: 

Line 13: typo, “iconic” should be “ionic”.

Line 195-199: what is the observation for the comparison between Figure 6(a) and 6(b)? Need to give more information on this.

Line 207-229 should be placed after line 188; 

Author Response

Response to Reviewer 2 Comments

Point 1: Line 13: typo, “iconic” should be “ionic”.

Response 1: The “iconic”has been modified as “ionic” in Line 13 and highlighted in blue.

Point 2: Line 195-199: what is the observation for the comparison between Figure 6(a) and 6(b)? Need to give more information on this.

Response 2: The comparison of Figure 6(a) and 6(b) and the comparative results have been provided and highlighted in blue.

Point 3: Line 207-229 should be placed after line 188.

Response 3: The Line 207-229 have been placed after line 188 and highlighted in blue.

Reviewer 3 Report

Tian et al. conducted a theoretical investigation of ANHARMONIC impact on the oxygen ion conductivity and thermal stability of 8% mol YSZ. I provided the following comments to criticize this manuscript. Hope the authors could clarify them and they are helpful  to further improve the quality of this manuscript:

1)      Develop the acronyms when you used them for the first time, such as YSZ. Authors should check the whole manuscript to do a better job

2)      When constructing the YSZ model, did author consider the oxygen vacancies?

3)      Adjust the format of Table 3. “Atoms” in a single line; there should be a space between the number value and units; what does that mean 102; 1012? Please use standard expressions. There are many format issues that need authors to be addressed;

4)      Please make the unit format consistent; for example. In Figure 5, it is T/K; while there is a , in Figure 6a; interestingly, there is no space between T and K-1. All of those are not consistent in the whole manuscript;

5)      The resolution of some figures is poor. Additionally, I need to significantly enlarge the figures to see their notations and values. The readability and quality of figures should be enhanced

6)      Authors need to add more introduction/knowledge of the thermal stability of YSZ. What is the thermal stability coefficient, and what does that mean regarding the materials aspect?

Can be improved

Author Response

Response to Reviewer 3 Comments

Point 1: Develop the acronyms when you used them for the first time, such as YSZ. Authors should check the whole manuscript to do a better job.

Response 1: We have carefully reviewed the entire manuscript and made revisions.

Point 2: When constructing the YSZ model, did author consider the oxygen vacancies?

Response 2: YSZ's high ionic conductivity mainly comes from its high concentration of oxygen vacancies. Therefore, this article considered the oxygen ion vacancy defects in modeling, as shown in Figure 2.

Point 3: Adjust the format of Table 3. “Atoms” in a single line; there should be a space between the number value and units; what does that mean 102; 1012? Please use standard expressions. There are many format issues that need authors to be addressed.

Response 3: We have standardized the format and made revisions throughout the manuscript.

Point 4: Please make the unit format consistent; for example. In Figure 5, it is T/K; while there is a , in Figure 6a; interestingly, there is no space between T and K-1. All of those are not consistent in the whole manuscript.

Response 4: We have modified the units format to be consistent throughout the manuscript.

Point 5: The resolution of some figures is poor. Additionally, I need to significantly enlarge the figures to see their notations and values. The readability and quality of figures should be enhanced.

Response 5: We have edited the figures to make them clearer.

Point 6: Authors need to add more introduction/knowledge of the thermal stability of YSZ. What is the thermal stability coefficient, and what does that mean regarding the materials aspect?

Response 6: The definition of thermal stability and the importance of researching thermal stability have been added in the text and highlighted in green.